# Elixhauser Comorbidity Measure and Charlson Comorbidity Index in Predicting the Death of Spanish Inpatients with Diabetes and Invasive Pneumococcal Disease

**DOI:** 10.3390/microorganisms13071642

**Published:** 2025-07-11

**Authors:** Enrique Gea-Izquierdo, Rossana Ruiz-Urbaez, Valentín Hernández-Barrera, Ángel Gil-de-Miguel

**Affiliations:** 1Department of Medical Specialties and Public Health, Rey Juan Carlos University, 28922 Madrid, Spain; 2Faculty of Medicine, Pontifical Catholic University of Ecuador, Quito 170143, Ecuador; 3María Zambrano Program, European Union, Madrid, Spain; 4Internal Medicine Service, Eugenio Espejo Hospital, Quito 170403, Ecuador; 5CIBER of Respiratory Diseases (CIBERES), Instituto de Salud Carlos III, 28029 Madrid, Spain

**Keywords:** diabetes, invasive pneumococcal disease, comorbidity index, retrospective study, Spain

## Abstract

Invasive pneumococcal disease (IPD) is a serious infection caused by the bacterium *Streptococcus pneumoniae* (pneumococcus) that can produce a wide spectrum of clinical manifestations. The aim of this study was to analyze the comorbidity factors that influenced the mortality in patients with diabetes (D) according to IPD. A retrospective study to analyze patients with D and IPD was carried out. Based on the discharge reports from the Spanish Minimum Basic Data Set (MBDS) from 1997 to 2022, the Elixhauser Comorbidity Index (ECI) and the Charlson Comorbidity Index (CCI) were calculated to predict in-hospital mortality (IHM) in Spain. A total of 12,994,304 patients with D were included, and 84,601 cases of IPD were identified. The average age for men was 70.23 years and for women 73.94 years. In all years, ECI and CCI were larger for type 2 D than for type 1 D, with men having a higher mean than women. An association was found between risk factors ECI, age, type 1 D, COVID-19, IPD (OR = 1.31; 95% CI: 1.29–1.35; *p* < 0.001); CCI, age, type 1 D, COVID-19, IPD (OR = 1.45; 95% CI: 1.42–1.49; *p* < 0.001), and increased mortality. The IHM increased steadily with the number of comorbidities and index scores from 1997 to 2022. D remains a relevant cause of hospitalization in Spain. Comorbidities reflected a great impact on patients with D and IPD, which would mean a higher risk of mortality. Predicting mortality events and length of stay by comparing indices showed that CCI outperforms ECI in predicting inpatient death after IPD.

## 1. Introduction

Diabetes (D) is a chronic disease that can seriously damage many organs and systems of the body, especially the blood vessels and nerves. The International Diabetes Federation estimated that 500 million adults aged 20–79 were living with D around the world in 2024, and its prevalence is expected to increase to 853 million people by the year 2050 [1]. Type 1 D is one of the most common chronic diseases in childhood and the major type of D in a pediatric population, but it can occur at any age [2].

D is steadily increasing in Western societies, and its positive association with the development of several types of infection is well-known [1,3,4]. On the other hand, common infections in patients with D may be caused by unusual pathogens, and the presentation and course may be different from those without D, which may delay the correct diagnosis and the adequate therapy [5,6,7]. The causative organisms of infections may differ from those identified in the general population, depending on the local nosocomial trends, glycemic control, and specific vulnerabilities generated by underlying comorbidities (e.g., D-related complications). Also, patients with D seem to be at particularly high risk of infection with certain bacterial microorganisms, such as *Streptococcus pneumoniae* (SP) (pneumococcus). Therefore, patients with D tend to be hospitalized for infections more frequently than non-diabetics [4,7], and comorbidities play a crucial role in this scenario.

SP is the most important bacterial respiratory pathogen in humans and causes a range of mucosal and invasive infections in both children and adults. In addition, it is a significant cause of morbidity and mortality worldwide [8]. Pneumococcal diseases are symptomatic infections caused by the bacterium SP, and, usually, the invasive pneumococcal disease (IPD) term is used for more severe and invasive pneumococcal infections [9]. Moreover, irrespective of age, the presence of certain underlying conditions is associated with an increased risk for IPD, including, but not limited to, D, asthma, chronic/cyanotic heart and lung diseases, diseases leading to immunosuppression, and chronic renal and liver diseases [10]. Despite data relating the risk of IPD to chronic underlying diseases, more research is needed about the prevalence of SP among the diabetic population and underlying risk groups.

Concerning the clinical practice of diabetes, and in relation to self-care, constant and precise support regarding behavioral education and dealing with patients to address the distress caused by the disease should be implemented [11]. Moreover, specific studies on lifestyle medicine nursing could determine a key role in the management of patients with type 2 diabetes or other chronic diseases [12], demonstrating that health education intervention is beneficial in controlling HbA1c among diabetics [13].

The primary objective of the study was to analyze the comorbidity factors that influenced the mortality in patients with D according to IPD, while the secondary one to assess the difference in the accuracy of the Elixhauser Comorbidity Index (ECI) and the Charlson Comorbidity Index (CCI) in predicting in-hospital mortality (IHM) after IPD.

## 2. Methods

### 2.1. Study Design

In this study, we retrospectively analyzed all patients requiring hospitalization for D in the Spanish National Health Service over 26 years. The population of interest was patients with newly diagnosed D. Patients with D were analyzed according to the presence of SP infection.

### 2.2. Data Source and Study Population

This study was conducted between January 1997 and December 2022, with approximately 98% of the hospitalized population covered. The hospital discharge report was obtained from the Spanish Government. Through the Minimum Basic Data Set (MBDS), belonging to the Ministry of Health (Government of Spain), patients with D (IPD/non-IPD) were obtained. Using the MBDS database, relevant conditions and comorbidities were identified using diagnostic and procedure codes. Weights were recorded for each discharge record, which were used in the analysis to obtain Spanish estimates.

Patients with a primary diagnosis of D were considered. Identification was carried out through the International Classification of Diseases (ICD), 9th Revision, Clinical Modification (ICD-9-CM), and 10th Revision, ICD-10-ES, codes (Appendix A) to screen for cases. The clinical criteria for case definition of IPD were associated with cases produced by the systemic dissemination of SP, ICD-9-CM (code 481), and ICD-10-ES (code J13). Influenza was identified by 487 and 488 codes (ICD-9-CM), and J11.1. code (ICD-10-ES); COVID-19 with U07.1. code (ICD-10-ES); and RSV disease with 079.6, 466.0, 466.1, 466.11, and 480.1 codes (ICD-9-CM), and B97.4, J12.1, J20.5, J20.9, J21.0, and J21.9 codes (ICD-10-ES).

The distribution of the population was determined using the global dependency index, with IHM (death from any cause during the hospitalization period) being the primary outcome of this study. Algorithms were developed to calculate the ECI and CCI from the discharge data of each patient.

### 2.3. Hospitalization and Demographic Data

Regarding hospital admissions, the Spanish Ministry of Health keeps clinical, demographic, and administrative data. With respect to hospital and medical service data, the ICD-9-CM and ICD-10-ES diagnostic codes were used, yielding an updated list of the categories employed. Hospitalization and demographic data were extracted from the electronic medical records of the MBDS, with a single diagnostic code per record. Demographic information included age and sex; clinical information included confirmed diagnosis of D, history of comorbidities conditions, hospital characteristics, inpatient outcomes (discharge disposition and hospital length of stay), developed complications during hospitalization, and mortality. Each information was retrieved from the database. The research used completely anonymous patient data, which were coded by specialized codifiers (using the ICD) based on medical records.

The diagnosis type was used to identify whether the diagnosis was considered a complication (a condition arising during the hospital stay) or a comorbidity (pre-existing condition). The patients’ characteristics as related to IPD, influenza, COVID-19, and RSV disease were analyzed. In addition to clinically relevant comorbidities included from baseline patient characteristics, Elixhauser and Charlson comorbidities were considered. Baseline characteristics were retrospectively collected, baseline medical history and comorbidities were administered, and the ECS and CCI were obtained.

### 2.4. Statistical Analysis

Descriptive statistics were used to summarize clinical data and risk factors. Categorical data were expressed as numbers (%) and continuous data as mean and standard deviation (SD).

Differences in variables between cases with/without IPD risk factor, and comorbidities potentially associated, were assessed. An amount of 30 comorbidity binary variables and 16 variables were used to calculate the Elixhauser and Charlson comorbidity indices, respectively. ECI was considered for use with comorbidities only, but there may be times when the index should consider all diagnoses (e.g., in a longitudinal study, all diagnoses could be included in the index calculation because they represent comorbidity over time). CCI predicted complications of the sum of certain diseases, such as functional capacity at discharge and mortality. In addition, it was created as a system for evaluating life expectancy at ten years, depending on the subject’s comorbidities and the age at which it is evaluated. The ECI was calculated according to the point system from van Walraven [14]. The Elixhauser weights range from −7 to +12, while the Charlson weights assigned to each comorbidity range from +1 to +6. Patients were divided into groups based on their comorbidities according to the risk status. We calculated the relative improvement in predictive performance of the Elixhauser score compared to the Charlson score. In addition, the weighted ECS, developed by van Walraven et al., was computed and further stratified into groups (<15, 15–39, 40–54, 55–64, 65–79, 70–79, and >80 years), respectively. The weights assigned to each comorbidity range from <0 to >5. Comorbidity scores can then be calculated for each patient by summing the individual weights of all comorbidities.

Variables indicating the presence or absence of each comorbidity were created, and their associations with mortality were assessed in bivariate analysis. Chi-square, Fisher’s exact test, and ANOVA were used to evaluate the relationship between the outcomes and the attributable factors. A multivariable analysis mortality model was performed using non-conditional logistic regression analysis, excluding collinearity and considering the confounding variables [15]. The Elixhauser and Charlson risk indices in all Spanish patients with D and IPD were analyzed. According to the scores obtained, different risk groups of mortality were determined. Crude mortality rates and group-specific age and sex were calculated by dividing the number of deaths by IPD by the number of IPD cases in each subgroup and were expressed as percentages. Multivariable logistic regression analyses were performed to assess the contributions of the individual comorbidities to predicted IHM. A comorbidity risk adjustment model regarding its ability to predict IHM was assessed. In all tests, the significance level used, *p* < 0.001, was considered statistically significant. Analyses were carried out using the Stata software version 16.1.

### 2.5. Ethical Statement

Because anonymous data were used and were obtained directly from the Spanish Ministry of Health, patient consent was not required to review their medical records.

## 3. Results

D episodes were diagnosed during the 26-year study period. The mean age was 71.89 years (SD 13.62), and a total of 12,944,304 cases with a diagnosis of IPD in the medical record were identified.

Of these patients included in the study, 509,096 (3.92%) had type 1 D and 12,485,208 (96.08%) had type 2 D (Figure 1). Of the patients with type 1 D, 260,321 were male, or 51.13% (mean age ± standard deviation, 44.78 ± 22.53 years), and 248,775 were female, or 48.57% (mean age ± standard deviation, 45.52 ± 24.59 years). According to the common age distribution of D1 in pediatric age, 66,237 (13.01%) children were aged <15 years. Of the patients with type 2 D, 6,916,624 were male, or 55.40% (mean age ± standard deviation, 71.19 ± 11.52 years), and 5,568,584 were female, or 44.60% (mean age ± standard deviation, 75.21 ± 11.85 years).

The relationship between CCI, ECI–van Walraven, and clinical parameters was analyzed (Table 1 and Table 2). Whereas type 1 D, COVID-19, and mortality were significantly different between patients in the IPD risk category (Table 3). IHM of the patients with D by age group in Spain, with/without IPD, is shown in Figure 2. Age was associated with mortality, which was significantly different between patients aged 65–79 years and >80 years. Significant differences were observed between age groups, where patients >80 years have higher index scores. Metastatic cancer, fluid and electrolyte disorders, weight loss, paralysis, and other neurological disorders were associated with IHM from most to least.

At least one D risk factor was present in 12,055,208 (92.80%) cases. Common risk factors included Hypertension (49.63%), Cardiac Arrhythmias (23.68%), COPD (18.66%), Congestive Heart Failure (18.60%), and Renal Failure (16.89%). Comorbidities were significantly more often male than female (2 ± 1.95 vs. 1.54 ± 1.7; *p* < 0.001 in CCI and 6.68 ± 7.28 vs. 5.5 ± 6.98; *p* < 0.001 in ECI), and more commonly at type 2 D compared with type 1 D (1.83 ± 1.87 vs. 1.10 ± 1.57; *p* < 0.001 in CCI and 6.29 ± 7.20 vs. 2.77 ± 5.33; *p* < 0.001 in ECI). There were differences between type 2 D and type 1 D in all comorbidities. Type 2 D had a Charlson comorbidity score of ≥3 more often than type 1 D (CCI ≥3 was 28.13% in type 2 versus 17.51 in type 1; *p* < 0.001). The Charlson Comorbidity score for the two groups was similar (difference 0.73). Hypertension and Cardiac Arrhythmias were the most common underlying conditions in patients with D. In general, all comorbidities were more frequent among patients with type 2 D than among type 1 D; only Drug Abuse and AIDS/HIV were less frequent. Type 2 D had an Elixhauser comorbidity score of ≥5 more often than type 1 D (ECI ≥ 5 was 52.56% in type 2 versus 29.90% in type 1; *p* < 0.001). Charlson Comorbidity scores were 0–2 in 72.3% and 3–≥5 in 27.74% and were higher in those with IPD risk factor (*p* < 0.001). Elixhauser Comorbidity scores were 0–4 in 48.33% and 51.67% in >5 cases, and they were also higher in those with IPD risk factor (*p* < 0.001). Detailed information regarding comorbidities in patients with an IPD diagnosis can be found in Table 4. Of the 84,601 cases with D meeting objective criteria for IPD, the prevalence of IPD risk factors was similar (0.44% type 1 D vs. 0.56% type 2 D). The male-to-female ratio was 1.60 (53,318/33,283). Patients with IPD risk factors were older (mean = 75.35 years vs. 71.86 years; *p* < 0.001).

The rates of IHM increased steadily with the number of comorbidities and index scores in multivariable logistic regression analysis. Influenza (OR = 0.64; 95% CI, 0.57–0.73; *p* < 0.001), COVID-19 (OR = 2.37; 95% CI, 2.34–2.41; *p* < 0.001), RSV disease (OR = 0.65; 95% CI, 0.63–0.66; *p* < 0.001), and IPD (OR = 1.45; 95% CI, 1.42–1.49; *p* < 0.001) had the highest odds of inpatient mortality in the Charlson algorithm.

Within the study population, overall IHM was 7% (914,910 of 12,994,304), with 10.25% in patients with IPD (8675 of 84,601). Type 1 and type 2 D IHMs were different between those with and without IPD. Type 2 D was more frequent in IPD (0.66% vs. 0.44%, *p* < 0.001). The IPD-attributable death in type 1 D patients was 9.47%, and among type 2 D patients was 10.28% (*p* < 0.001). Readmission and length of hospital stay (Table 2) were not found to be associated with any attributable factors. Length of hospital stay was significantly different only between patients in infection/non-infection categories (11.22 vs. 9.11; *p* < 0.001).

## 4. Discussion

D influences the outcome of specific infections, such as bacteremia and mortality following IPD. In fact, there is a higher mortality rate among patients with D, which is partly attributable to IPD. It is logical to see that D can be present in older adults who have IPD, and most likely, this is a product of age, together with the age-dependent trend in the incidence of IPD. Additionally, the rate of SP infections in adults is thought to be increasing in recent years [16,17,18], and pneumococcus is usually associated with some seasonal respiratory viruses such as respiratory syncytial virus and influenza, a virus that has also been suggested to induce type 1 D [19,20]. To this, we must add that some recent studies have suggested that the COVID-19 pandemic was associated with increased D risk [21,22].

An evaluation of the sociodemographic distribution and clinical patterns associated with patients with D requiring hospitalization and their impact on mortality was carried out. An observational multicenter study to estimate the potential of IPD in the IHM related to D patients was performed. Using all IPD cases admitted during 1997–2022, we created a risk model generating outcome estimates on the risk of dying for each patient. This study demonstrated that the prevalence of SP in D patients with IPD was elevated, higher in males, and more frequent in elderly patients.

Firstly, the study identifies that the mean age was significantly higher in patients with type 2 D than in type 1 D, without significant heterogeneity among groups. These results are consistent with the clinical experience that elderly patients with D are particularly prone to infection, and senescence of the immune system can also alter host defense mechanisms. Moreover, the prevalence of D has been reported to increase in older groups in the general population. Second, several studies have found that patients with D spend longer periods of time in the hospital than those without. A higher prevalence of extended space infection in patients with D also leads to a high frequency of complications, including pneumonia, pericarditis, hypoproteinemia, skin defect, airway obstruction, pleural effusion, mediastinitis, intracranial infection, diabetic ketoacidosis, and mortality. A significant prevalence of these complications in patients with D (RR = 2.43; 95% CI, 1.80–3.30) compared to those without was found [23].

Infections in patients with type 1 D are significantly more common and severe than in healthy subjects, and infections cause several medical, social, and economic problems [24,25]. Given the fact that SP is the cause of IPD in diabetics and that pneumococcal infections are potentially vaccine-preventable, it is worth recommending such vaccination in people with D. Particular attention should be paid to ensuring that all patients with D receive both pneumococcal conjugate vaccines (PCVs) and polysaccharide vaccines, as these individuals are highly susceptible to infection with encapsulated bacteria [26].

European countries raise IPD as a priority research topic and the impact of pneumococcal vaccination on the prevention of serious diseases in current vaccination programs in the European Union and the European Economic Area (considering the new vaccines 15-valent, 20-valent, serotypes, cross protection, mixed vaccination schedules, target populations, and serotype replacement) [27]. In Spain, a decreased risk of IPD was observed in all age groups in recent years, not identified before the introduction of PCVs in childhood vaccination programs [28]. For this country, in the age group of >65 years, the incidence rate per 100,000 inhabitants in 2022 (14.72) doubled that of the previous year (7.69), although it continued to be lower than the pre-pandemic incidence rate. The greater use of non-pharmacological measures, if we compare it with the pre-pandemic era, could explain this event, but also the increase in flu coverage in recent years, and probably the increase in vaccination coverage against pneumococcus in this age group. In recent years, in the older age groups (15–44 years, 45–64 years, and ≥65 years), after cases due to vaccine serotypes 3 and 8, cases due to vaccine serotypes 9N and 22F follow in frequency. Serotype 8, in the three age groups, is the one in which the greatest increase is observed from 2015 to 2019. The most frequent cases due to non-vaccine serotypes (not included in any vaccine) in the three age groups were the serotypes 35F, 23A, 31, 6C, 24F, 23B, 35B, 16F, and 15A. Overall, except for serotype 3, the rest of the cases due to serotypes included in the PCV13 vaccine have been decreasing in older age groups [29,30]. In the aforementioned period, in the three age groups, a decrease in the number of cases due to serotypes included in the PCV13 vaccine was observed. On the contrary, no reduction was observed in the number of cases due to serotypes included in the PCV20 and PPSV23 vaccines (a consequence of the increase in serotypes included in PCV20 or PPSV23 and not included in PCV13). During the pandemic years 2020 and 2021, there was a decrease in cases due to vaccine and non-vaccine serotypes [29]. In the 65 and older age group, 70.2% of IPDs for which the serotype was reported were caused by a serotype included in the pneumococcal polysaccharide vaccine (PPV23). However, the lack of knowledge of vaccination coverage in this population (probably low) with the PPSV23 vaccine makes it impossible to analyze its effectiveness. A total amount of 36.8% and 64.8% of IPD were caused by the serotypes included in the PCV15 and PCV20 vaccines [31], respectively, recently authorized in Spain.

In Spain, vaccination coverage has remained high in recent years, and, therefore, the increase in IPD cases cannot be explained by a decrease in vaccination coverage. The increase could be due to less exposure to the circulation of pneumococcal strains and, therefore, natural immune reinforcements during the first years of the pandemic. In fact, vaccination against SP is the most effective measure to prevent IPD. In this country, it is part of the common lifelong vaccination schedule, and, since 2019, it has been systematically administered to the adult population aged 65 and over. However, vaccination status data is not available in most cases, making its analysis impossible, but the primary vaccination coverage published by the Ministry of Health since 2017 is greater than 97% [32]. Together with the need to increase vaccine uptake in all children aged <2 years, PCV booster doses in patients with D (type 1) are needed to maintain the protection offered by the vaccines.

Furthermore, in the study, we identified that from an Elixhauser comorbidity score of >0, there is an increase in mortality of 26%, increasing to 86% if we consider the score to be more than 4. Starting from 5 or more, the mortality rate is 4.14 times higher than those who had the lowest score in the Elixhauser comorbidity score. High-risk patients with a score >5 or more should be carefully considered in patients with D. The prognosis should also be discussed with the patient to achieve shared decision making. The analysis of specific comorbidities (Influenza, RSV, IPD) confirmed upward trends in patient proportions with IPD who were admitted with these comorbidities. Moreover, we found that patients with D had more comorbid conditions, and this may have had an impact on the etiology and prognosis of IPD, with Hypertension being the most frequent. The factors Metastatic Cancer and Congestive Heart Failure were known predictors of mortality, but Dementia has not been thoroughly investigated for its risk of mortality, but could reflect a patient’s condition and impaired prognosis. Limiting the analysis to predict mortality only to the clinical variables, ECI was inferior to the CCI.

In Spain, starting in 2016, a downward trend in fatality was observed in the older age groups (45–64 years and ≥65 years). This data coincides with the changes introduced in the vaccination program; however, as few Autonomous Communities have reported death data, the analysis cannot be conclusive. For this reason, it is considered that continuous and quality epidemiological surveillance of IPD is necessary, with a systematic collection of clinical–epidemiological data, vaccination history, and laboratory data to be able to adequately assess the evolution of the incidence of the disease and the impact of vaccination. Surveillance of the disease burden due to vaccine serotypes in Spain is essential, which will allow the review of strategies to achieve optimal protection in the elderly population. The available evidence on the safety and immunogenicity of the PCV15 and PCV20 vaccines is promising, but additional data will need to be provided on the effectiveness of the vaccines [29], the duration of the immune response, and the replacement phenomenon after the introduction of these new vaccines. Additionally, it is necessary to disseminate and remind health professionals about the new vaccination recommendations against pneumococcus [33] in patients with D.

Regarding quality, it should be noted that in 2022, all the Autonomous Communities reported cases of IPD, which allows to have a better image of the epidemiology of this disease in Spain. In addition, the degree of completion of essential variables such as vaccination status improved compared to previous years, which had low exhaustivity. However, the small number of cases in which vaccination has been reported has made its evaluation impossible. It is estimated that this information is essential for the continuous evaluation of the vaccination program against IPD, especially after the recent authorization of new vaccines. Only quality data will support the public health response and vaccination policy [34].

Education is considered a fundamental part of diabetic patient care. People with diabetes, whether or not they use insulin, must assume responsibility for the daily management of their disease. Therefore, it is essential that they understand the disease and know how to treat it. The goal of educating people with diabetes is to improve knowledge and skills and integrate self-management into daily life. Sometimes, it is preferable to focus on the effectiveness of self-management, rather than education or specific educational components [35].

Our study has several strengths. We had a large sample size to analyze the epidemiology of IPD in patients with D. The study period of 26 years included both pre- and post-PCV13 era, presenting a very reliable epidemiological view.

In summary, this study demonstrated that patients with D are frequently colonized by SP, and the resulting impact on mortality, the need to investigate the prevalence of SP carriage, and serotype distribution in the diabetic population with well-defined high-risk conditions for IPD. Append that there is a lack of sufficient data regarding SP carriage in patients with D and chronic diseases, and there are limited data on the incidence of IPD in individuals with underlying medical conditions. Given these potential risks, we believe that further research is needed to understand IPD cases with D risk factors.

## 5. Conclusions

Health general networks are useful ways for epidemiological surveillance of diseases such as IPD. This study confirms electronic database (MBDS) as a reliable epidemiological tool for estimating the mortality of IPD in patients with D. It provides an important source of information on the disease, which can be useful to define trends pre- and post-vaccines’ introduction and the improvement of clinical practice.

In conclusion, IPD remains a highly lethal complication in patients with D. Our report shows that the comorbidities/IPD in type 1 D differ from type 2 D. Elderly patients and the high proportion of major comorbidities are risk factors for dying from IPD in a hospital. Predicting mortality events and length of stay by comparing indexes shows that the CCI outperforms ECI in predicting inpatient death after IPD.

## 6. Limitations

We did not have the information on the immunization status of patients. Inclusion of vaccination status could provide a better understanding of the effectiveness of PCV13 in preventing serotype 3 infection. In addition, it must be considered the possible generalization of the collected data, the type of study conducted, and the tool adopted. The long study period may integrate changes in the modern comorbidities’ therapies, lack of follow-up of patients, the introduction of vaccines in the population, in the antimicrobial resistance or medical procedures, readmission of patients with D and IPD, the non-inclusion of some private clinics, possible changes in the coding of comorbidities, and risk factors not included in MBDS. Further to what has been mentioned, it should be added that in retrospective data collection, early cases could be missed. To investigate whether missing data might influence our results, it would be pertinent to create scenarios in which all prevalent cases (with an event before the date of the first visit to the Spanish Health Service) were missed, thereby excluding the prevalent cases from the analysis and considering only incident cases. Finally, it would be appropriate to analyze the differences in the prevalence of some diagnoses to identify possible underdiagnoses.

## Figures and Tables

**Figure 1 microorganisms-13-01642-f001:**
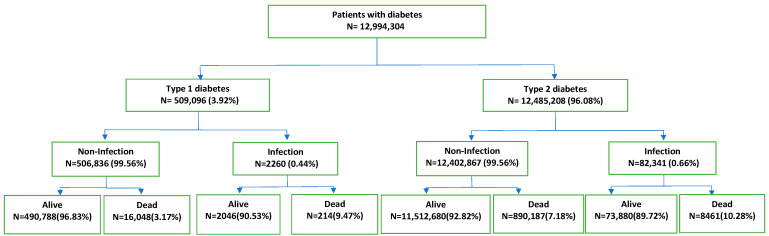
Flowchart of patients with diabetes.

**Figure 2 microorganisms-13-01642-f002:**
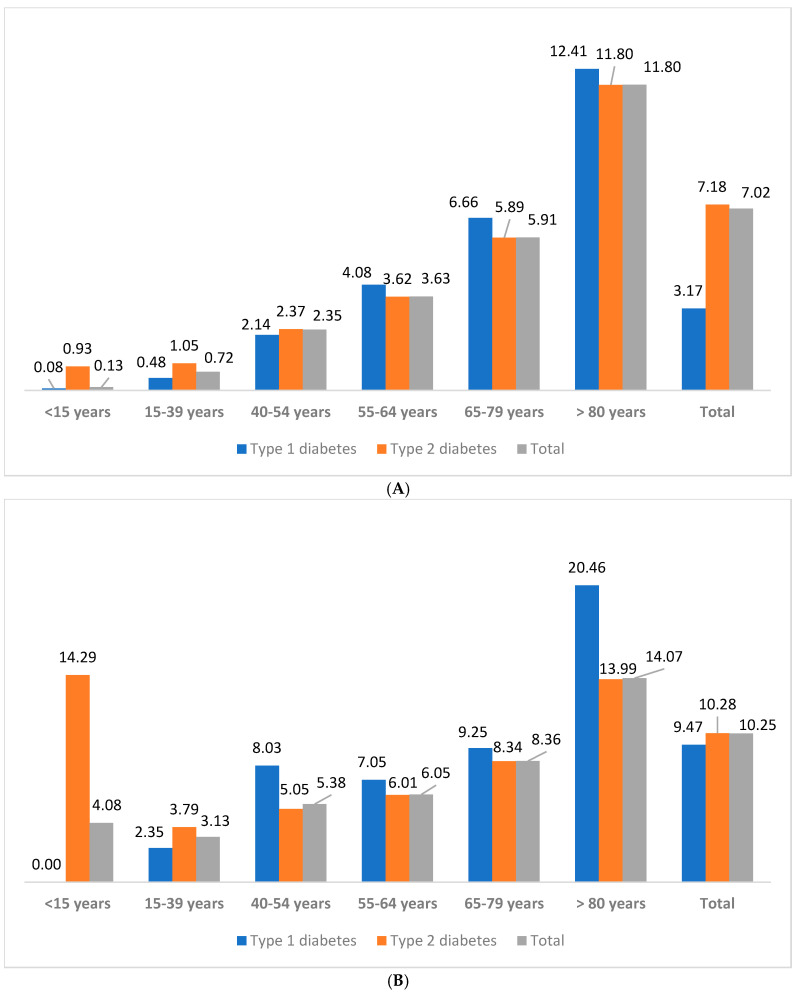
IHM (%) of the patients with diabetes by age group in Spain, from 1997 to 2022. (**A**) With invasive pneumococcal disease. (**B**) Without invasive pneumococcal disease.

**Table 1 microorganisms-13-01642-t001:** Relationships between Charlson Comorbidity Index, Elixhauser Comorbidity Index–van Walraven, and clinical parameters.

		Charlson Comorbidity Index	Elixhauser Comorbidity Index–van Walraven
		(Mean ± SD)	(Mean ± SD)
Sex	Male	2 ± 1.95	6.68 ± 7.28
	Female	1.54 ± 1.7	5.5 ± 6.98
Age group	<15 years	0.06 ± 0.34	0.62 ± 2.15
	15–39 years	0.58 ± 1.21	1.03 ± 4.33
	40–54 years	1.33 ± 1.77	3.27 ± 6.38
	55–64 years	1.68 ± 1.92	4.82 ± 6.97
	65–79 years	1.87 ± 1.9	6.16 ± 7.12
	>80 years	1.97 ± 1.78	7.84 ± 7.16
Diabetes	Type 1	1.1 ± 1.57	2.77 ± 5.33
	Type 2	1.83 ± 1.87	6.29 ± 7.20
Invasive pneumococcal disease	No	1.8 ± 1.86	6.15 ± 7.17
	Yes	1.65 ± 1.64	6.65 ± 6.78
Influenza	No	1.8 ± 1.86	6.15 ± 7.17
	Yes	1.94 ± 1.77	7.36 ± 7.32
COVID-19	No	1.8 ± 1.86	6.15 ± 7.16
	Yes	1.81 ± 1.9	6.61 ± 7.61
RSV disease	No	1.8 ± 1.86	6.13 ± 7.17
	Yes	1.8 ± 1.64	7.30 ± 7.05
Death	No	1.71 ± 1.8	5.79 ± 6.96
	Yes	2.93 ± 2.28	10.92 ± 8.1
Total		1.8 ± 1.86	6.15 ± 7.17

**Table 2 microorganisms-13-01642-t002:** Relationships between the Charlson Comorbidity Index (0 vs. ≥5), Elixhauser Comorbidity Index–van Walraven (<0 vs. ≥5), and clinical parameters.

		Charlson Comorbidity Index					Elixhauser Comorbidity Index–van Walraven				
		0	1–2	3–4	≥5	*p*-Value	<0	0	1–4	≥5	*p*-Value
Sex	Male	1,775,916 (47.28)	3,130,806 (55.53)	1,476,444 (60.87)	793,779 (67.54)	<0.001	432,937 (40.89)	1,707,271 (52.14)	1,171,608 (60.19)	3,865,129 (57.56)	<0.001
	Female	1,979,948 (52.72)	2,506,898 (44.47)	948,966 (39.13)	381,547 (32.46)		625,730 (59.11)	1,567,042 (47.86)	774,932 (39.81)	2,849,655 (42.44)	
Age		67.43 (16.83)	73.31 (11.85)	74.93 (11.32)	73.07 (10.68)	<0.001	66.99 (13.97)	66.94 (16.58)	72.13 (11.27)	75.01 (11.49)	<0.001
Age group	<15 years	67,660 (1.8)	2767 (0.05)	210 (0.01)	35 (0)	<0.001	1625 (0.15)	60,205 (1.84)	1763 (0.09)	7079 (0.11)	<0.001
	15–39 years	190,800 (5.08)	55,787 (0.99)	17,749 (0.73)	4524 (0.38)		40,782 (3.85)	163,244 (4.99)	16,633 (0.85)	48,201 (0.72)	
	40–54 years	386,132 (10.28)	334,944 (5.94)	109,867 (4.53)	57,694 (4.91)		147,751 (13.96)	336,096 (10.26)	112,360 (5.77)	292,430 (4.36)	
	55–64 years	636,918 (16.96)	785,722 (13.94)	284,115 (11.71)	179,169 (15.24)		215,469 (20.35)	597,412 (18.25)	306,030 (15.72)	767,013 (11.42)	
	65–79 years	1,600,847 (42.62)	2,586,455 (45.88)	1,073,899 (44.28)	584,066 (49.69)		456,636 (43.13)	1,426,535 (43.57)	985,208 (50.61)	2,976,888 (44.33)	
	>80 years	873,507 (23.26)	1,872,029 (33.21)	939,570 (38.74)	349,838 (29.77)		196,404 (18.55)	690,821 (21.1)	524,546 (26.95)	2,623,173 (39.07)	
Diabetes	Type 1	267,836 (7.13)	152,104 (2.7)	68,515 (2.82)	20,641 (1.76)	<0.001	42,446 (4.01)	263,484 (8.05)	50,939 (2.62)	152,227 (2.27)	<0.001
	Type 2	3,488,028 (92.87)	5,485,600 (97.3)	2,356,895 (97.18)	1,154,685 (98.24)		1,016,221 (95.99)	3,010,829 (91.95)	1,895,601 (97.38)	6,562,557 (97.73)	
Invasive pneumococcal disease	No	3,734,164 (99.42)	5,594,993 (99.24)	2,410,232 (99.37)	1,170,314 (99.57)	<0.001	1,052,976 (99.46)	3,258,325 (99.51)	1,932,025 (99.25)	6,666,377 (99.28)	<0.001
	Yes	21,700 (0.58)	42,711 (0.76)	15,178 (0.63)	5012 (0.43)		5691 (0.54)	15,988 (0.49)	14,515 (0.75)	48,407 (0.72)	
Influenza	No	3,754,706 (99.97)	5,635,406 (99.96)	2,424,164 (99.95)	1,174,914 (99.96)	<0.001	1,058,234 (99.96)	3,273,541 (99.98)	1,945,738 (99.96)	6,711,677 (99.95)	<0.001
	Yes	1158 (0.03)	2298 (0.04)	1246 (0.05)	412 (0.04)		433 (0.04)	772 (0.02)	802 (0.04)	3107 (0.05)	
COVID-19	No	3,705,585 (98.66)	5,579,624 (98.97)	2,391,769 (98.61)	1,160,705 (98.76)	<0.001	1,042,217 (98.45)	3,242,210 (99.02)	1,926,470 (98.97)	6,626,786 (98.69)	<0.001
	Yes	50,279 (1.34)	58,080 (1.03)	33,641 (1.39)	14,621 (1.24)		16,450 (1.55)	32,103 (0.98)	20,070 (1.03)	87,998 (1.31)	
RSV disease	No	3,709,625 (98.77)	5,530,121 (98.09)	2,380,573 (98.15)	1,161,330 (98.81)	<0.001	1,042,097 (98.43)	3,241,196 (98.99)	1,915,798 (98.42)	6,582,558 (98.03)	<0.001
	Yes	46,239 (1.23)	107,583 (1.91)	44,837 (1.85)	13,996 (1.19)		16,570 (1.57)	33,117 (1.01)	30,742 (1.58)	132,226 (1.97)	
Death	No	3,657,311 (97.38)	5,265,970 (93.41)	2,184,598 (90.07)	971,515 (82.66)	<0.001	1,035,236 (97.79)	3,182,080 (97.18)	1,859,237 (95.51)	6,002,841 (89.4)	<0.001
	Yes	98,553 (2.62)	371,734 (6.59)	240,812 (9.93)	203,811 (17.34)		23,431 (2.21)	92,233 (2.82)	87,303 (4.49)	711,943 (10.6)	
Hospital stay; median (IQR)		5 (7)	7 (7)	7 (8)	8 (10)	<0.001	6 (7)	5 (7)	6 (8)	7 (8)	<0.001

**Table 3 microorganisms-13-01642-t003:** Risk factors for IHM in patients without/with IPD.

		IHM Charlson Predictor						IHM Elixhauser Predictor						
		Without IPD		With IPD		Total			Without IPD		With IPD		Total	
		OR (CI 95%)	*p*-Value	OR (CI 95%)	*p*-Value	OR (CI 95%)	*p*-Value		OR (CI 95%)	*p*-Value	OR (CI 95%)	*p*-Value	OR (CI 95%)	*p*-Value
Scores	0	1		1				<0	1		1		1	
	1–2	2.28 (2.26–2.3)	0.000	1.47 (1.38–1.56)	0.000	2.27 (2.25–2.28)	0.000	0	1.26 (1.24–1.27)	0.000	1.7 (1.47–1.96)	0.000	1.26 (1.24–1.28)	0.000
	3–4	3.38 (3.35–3.41)	0.000	1.92 (1.79–2.06)	0.000	3.36 (3.33–3.39)	0.000	1–4	1.81 (1.78–1.84)	0.000	1.43 (1.24–1.66)	0.000	1.81 (1.78–1.83)	0.000
	≥5	7.2 (7.14–7.25)	0.000	3.15 (2.88–3.45)	0.000	7.15 (7.09–7.2)	0.000	≥5	4.16 (4.1–4.21)	0.000	2.99 (2.62–3.42)	0.000	4.14 (4.09–4.2)	0.000
Sex	Male	1		1		1		Male	1		1		1	
	Female	1.09 (1.08–1.09)	0.000	1.06 (1.01–1.11)	0.012	1.09 (1.08–1.09)	0.000	Female	1.01 (1.01–1.02)	0.000	1.02 (0.97–1.07)	0.472	1.01 (1.01–1.02)	0.000
Age group	<15 years	1		1		1		<15 years	1		1		1	
	15–39 years	3.97 (3.22–4.89)	0.000	0.69 (0.16–3.01)	0.619	3.93 (3.19–4.83)	0.000	15–39 years	5.05 (4.1–6.23)	0.000	0.81 (0.19–3.57)	0.786	5 (4.06–6.15)	0.000
	40–54 years	9.82 (8–12.06)	0.000	1.22 (0.29–5.07)	0.787	9.72 (7.93–11.91)	0.000	40–54 years	14.08 (11.47–17.29)	0.000	1.49 (0.36–6.2)	0.585	13.91 (11.35–17.04)	0.000
	55–64 years	13.7 (11.16–16.82)	0.000	1.38 (0.33–5.72)	0.657	13.53 (11.05–16.58)	0.000	55–64 years	20.02 (16.31–24.59)	0.000	1.65 (0.4–6.88)	0.488	19.74 (16.11–24.18)	0.000
	65–79 years	21.69 (17.67–26.63)	0.000	1.9 (0.46–7.87)	0.376	21.41 (17.48–26.23)	0.000	65–79 years	29.91 (24.36–36.72)	0.000	2.2 (0.53–9.12)	0.277	29.46 (24.05–36.09)	0.000
	>80 years	45.12 (36.75–55.39)	0.000	3.41 (0.82–14.11)	0.091	44.47 (36.3–54.48)	0.000	>80 years	56.1 (45.69–68.88)	0.000	3.72 (0.9–15.41)	0.071	55.22 (45.08–67.65)	0.000
Diabetes	Type 2	1		1		1		Type 2	1		1		1	
	Type 1	1.09 (1.07–1.11)	0.000	1.32 (1.14–1.53)	0.000	1.1 (1.08–1.11)	0.000	Type 1	1.12 (1.1–1.14)	0.000	1.34 (1.15–1.56)	0.000	1.12 (1.1–1.14)	0.000
Influenza		0.65 (0.57–0.73)	0.000	0.24 (0.03–1.77)	0.163	0.64 (0.57–0.73)	0.000		0.63 (0.55–0.71)	0.000	0.24 (0.03–1.79)	0.165	0.62 (0.55–0.71)	0.000
COVID-19		2.38 (2.34–2.41)	0.000	1.98 (1.69–2.32)	0.000	2.37 (2.34–2.41)	0.000		2.29 (2.26–2.33)	0.000	2 (1.71–2.34)	0.000	2.29 (2.26–2.32)	0.000
RSV disease		0.65 (0.63–0.66)	0.000	0.86 (0.62–1.2)	0.373	0.65 (0.63–0.66)	0.000		0.6 (0.59–0.61)	0.000	0.86 (0.62–1.2)	0.384	0.6 (0.59–0.61)	0.000
Invasive pneumococcal disease						1.45 (1.42–1.49)	0.000						1.31 (1.29–1.35)	0.000

**Table 4 microorganisms-13-01642-t004:** Comorbidities in patients with IPD diagnosis.

Charlson Comorbidities	n	%	Elixhauser Comorbidities	n	%
COPD (Chronic Obstructive Pulmonary Disease)	2,424,418	18.66	Hypertension, uncomplicated	6,448,483	49.63
CHF (Congestive Heart Failure)	2,416,680	18.60	Cardiac Arrhythmias	3,076,757	23.68
RD (Renal Disease)	2,201,111	16.94	Chronic Pulmonary Disease	2,424,418	18.66
CeVD (Cerebrovascular Disease)	1,377,078	10.60	CHF (Congestive Heart Failure)	2,416,680	18.60
Cancer	1,199,085	9.23	Hypertension, complicated	2,292,632	17.64
AMI (Acute Myocardial Infarction)	1,137,390	8.75	Renal Failure	2,194,870	16.89
PVD (Peripheral Vascular Disease)	1,124,810	8.66	Obesity	1,591,944	12.25
Mild Liver Disease	687,840	5.29	Solid Tumor Without Metastasis	1,455,682	11.20
Dementia	604,624	4.65	Valvular Disease	1,231,107	9.47
Metastatic Cancer	570,768	4.39	Peripheral Vascular Disorders	1,124,810	8.66
Moderate/Severe LD (Liver Disease)	279,339	2.15	Liver Disease	974,433	7.50
Rheumatoid Disease	208,252	1.60	Fluid and Electrolyte Disorders	854,057	6.57
HP/PAPL (Hemiplegia or Paraplegia)	191,295	1.47	Alcohol Abuse	742,323	5.71
PUD (Peptic Ulcer Disease)	144,260	1.11	Other Neurological Disorders	737,802	5.68
AIDS/HIV	23,334	0.18	Depression	711,543	5.48
			Hypothyroidism	704,803	5.42
			Deficiency Anemia	643,158	4.95
			Pulmonary Circulation Disorders	574,695	4.42
			Metastatic Cancer	570,768	4.39
			Rheumatoid Arthritis/Collagen Vascular Diseases	257,013	1.98
			Weight Loss	234,095	1.80
			Coagulopathy	210,486	1.62
			Paralysis	191,295	1.47
			Blood Loss Anemia	147,196	1.13
			Lymphoma	130,278	1.00
			Psychoses	122,676	0.94
			Peptic Ulcer Disease Excluding Bleeding	80,487	0.62
			Drug Abuse	64,548	0.50
			AIDS/HIV	23,334	0.18

## Data Availability

The data sets analyzed in the current study are publicly available in the Hospital Discharge Records in the Spanish National Health System (MBDS) repository at https://www.mscbs.gob.es/en/estadEstudios/estadisticas/cmbdhome.htm (accessed on 26 January 2024). The information contained in this repository can be accessed without the need for any administrative permissions.

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
