# Peer review of "Elixhauser Comorbidity Measure and Charlson Comorbidity Index in Predicting the Death of Spanish Inpatients with Diabetes and Invasive Pneumococcal Disease"

_microorganisms, 2025, doi:10.3390/microorganisms13071642_

Round 1

Reviewer 1 Report

Comments and Suggestions for Authors

Dear Authors,

First of all, I would like to express my sincere gratitude for the opportunity to contribute my opinion to the evaluation of your manuscript. I found the topic interesting and relevant to our field.

Below, I list the main areas that could benefit from further elaboration and revision. Editing: I suggest that the authors, in line with recent field guidelines (see: https://diabetesjournals. org/care/issue/48/Supplement_1), use the term T2D instead of T2DM and Diabetes instead of Diabetes Mellitus. Attention of the acronym use, not always correct: I suggest a general check.I suggest the use of "gender" and not "sex".

Title: I suggest to reduce in max 15/20 words and insert the type of study conducted (see above for “mellitus”)

Abstract: an expansion of this section is needed in terms of “Clinical Practice” (see next suggestion).

Keywords: I suggest 4/5 that relevant for principal topic, population and type of study conduct as the correct title: see above for “mellitus”

Introduction: This lack of citation undermines the credibility of the statements made and should be addressed. Missing the Clinical practice view and, for this reason, I suggest the authors to extend the topics: “Lifestyle Medicine Case Manager Nurses for Type Two Diabetes Patients”, “Effect of community-based nurse-led support intervention in the reduction of HbA1c levels”, and “Relationship Among Diabetes Distress, Health Literacy, Diabetes Education, Patient-Provider Communication and Diabetes Self-Care”, which will certainly broaden the audience of potential readers and researchers interested in the data collected, which, moreover, have a significant impact on multidisciplinary view and full complete the section. For the objective, I recommend adopting the classic structure: “The primary objectives of the study were… while the secondary ones…” and eventually the research question.

Methods: This section is, in my view, the weakest and definitely warrants more attention. Several elements are either missing or insufficiently described. The reporting tool is absent, as is the related checklist in the supplementary files and the corresponding reference depending on the study design (see: https://www.equator-network.org/). Using such a tool would help the authors to present this crucial section more clearly and comprehensively.

Results: Overall, this section is well done and arguably the strength of the study. It will certainly benefit from the previous and following suggestions. Please improve the vision of the flow of the study; not clear.

Discussion: I would suggest focusing more on clinical practice implications and “Perspectives for Clinical Practice” section could be add and suppose to extend the management of care of this patients in several point of view. You could also summarize the collected data in this light, based on the earlier suggestions (introduction).

Limitations: In my opinion, these should be addressed in the possible generalization of the collected data, the type of study conducted and tool adopted. Specific section I suggest to adopt.

Conclusions: I recommend developing a critical analysis here as well, following the considerations previously outlined.

References: see the comments above and update the reference over 10 years if not for method section or relevant evidence based data finding.

In summary, I suggest responding point by point to each individual suggestion for possible reconsideration.

Reviewer 2 Report

Comments and Suggestions for Authors

The manuscript presented is interesting and is based on a large number of study subjects. The results obtained are good, but the manuscript also has some weaknesses, which I list below.
1.    When analyzing patients with diabetes, I believe it is important that the authors have included glycosylated hemoglobin and glucose data, as these two values greatly influence the comorbidities of patients. Similarly, it is not specified whether the patients are vaccinated. In addition, other variables such as smoking, obesity, etc. are not specified. 
2.    The statistical methods are not clearly specified. For example, the variables included in the regression study are not mentioned, nor is it specified whether collinearity was present, among other points.
3.    Some of the tables presented are incomplete.
4.    The limitations of the study need to be expanded, for example, the non-inclusion of private clinics or possible changes in the coding of comorbidities.
5.    The methodology section is very similar to a previous work by the authors (https://doi.org/10.3390/pathogens14050411).

Round 2

Reviewer 1 Report

Comments and Suggestions for Authors

Dear Authors,

very good job. In this form ready for publication.

Best

Reviewer 2 Report

Comments and Suggestions for Authors

I agree with the changes made by the authors.